# SUBSET SELECTION-BASED ATTRIBUTION REGULARIZATION FOR RATIONAL AND STABLE INTERPRETABILITY

## ABSTRACT

While explainable AI (XAI) has developed numerous attribution mechanisms to enhance model transparency, existing post-hoc methods remain limited to improving attribution faithfulness. In contrast, attribution invariance and rationality stem from the model's internal parameters, requiring specialized constraints during training for improvement. Current training strategies, on the one hand, supervised rationality-enhancing methods depend on manual annotations that incorporate human priors may conflict with the model's intrinsic decision reasoning. On the other hand, self-supervised invariance regularization methods rely on gradient-based attribution methods (e.g., Grad-CAM) with low faithfulness, resulting in misaligned explanations with the actual logic. Such not only hinders attribution refinement but also adversely affects task performance. To overcome these challenges, we introduce a training framework grounded in high-faithfulness submodular attribution, which enables the extraction of compact, discriminative pseudo-ground-truth regions without manual supervision. By integrating spatial constraints and high-confidence sample filtering, our approach effectively suppresses irrelevant areas and supplies high-quality attribution targets that positively guide model training. However, submodular attribution encounters non-differentiable and path-dependent issues of black-box attribution searches. We propose a novel submodular ranking loss that enforces search path consistency and termination alignment under geometric transformations, enabling differentiable optimization of the greedy search process. Extensive evaluation across classification accuracy, attribution stability, faithfulness, rationality, and precision shows that our method significantly enhances attribution quality with minimal effect on task performance.

## 1 INTRODUCTION

The rapid advances and successes of deep learning technologies in domains such as image recognition and natural language processing heavily rely on increasingly complex and high-dimensional parameter configurations. However, this expansion has also led to growing challenges regarding model explainability, which significantly limits the broad application of deep learning in high-stakes fields, such as medical diagnosis (Tjoa & Guan, 2021), legal decision-making (Vale et al., 2022), and autonomous driving (Kuznietsov et al., 2024; Nie et al., 2024), where the reliability and interpretability of decisions are critically required.

A considerable number of studies in Explainable Artificial Intelligence (XAI) (Zhao et al., 2024; Sundararajan et al., 2017; Zhuo & Ge, 2024; Zhang et al., 2024) aim to enhance model transparency and interpretability by developing more sophisticated attribution techniques, thereby rendering the decision-making process more comprehensible to humans. However, existing attribution mechanisms predominantly focus on improving attribution faithfulness (Chen et al., 2025), which evaluates whether an explanation accurately reflects the true causal relationship between the input data and the model's inference outcome. The lack of robustness and semantic rational in attributions stems from the weak or unstable association between the features learned by the model and the discriminative semantic cues (Karimi et al., 2023). For instance, when geometric transformations (Pillai et al., 2022; Crabbé & van der Schaar, 2023) are applied to the input, the resulting attribution may exhibit significant inconsistency (see Fig. 1), suggesting that the model fails to capture semantically invari-

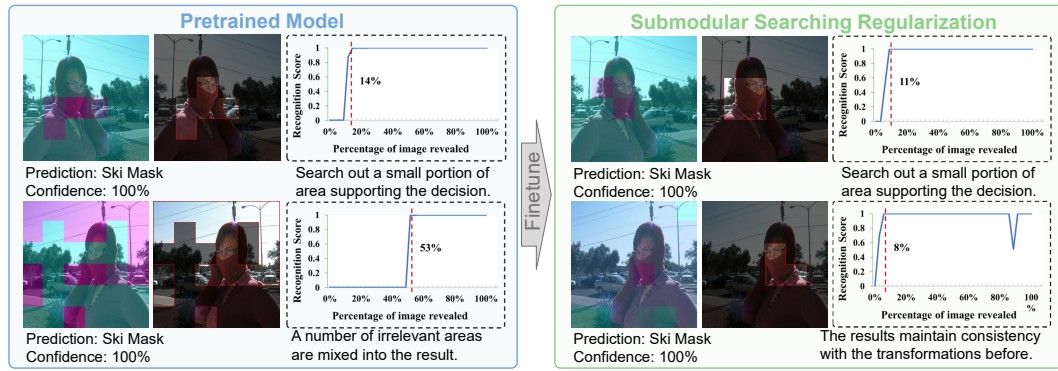

Figure 1: Comparison of attribution between the pre-trained and the regularized model. We visualize the attribution maps obtained via submodular search, along with the corresponding recognition heatmaps and Insertion AUC curves, before and after flipping of the input image. Both the pre-trained and regularized models can correctly identify the object with high confidence. However, the pre-trained model exhibits inconsistent focus regions after flipping, whereas the regularized model maintains perceptual consistency under transformation. Moreover, the regularized model produces more compact attribution regions and achieves higher Insertion AUC scores.

ant representations under transformation (Ruan et al.). This highlights the necessity of introducing explicit supervision and regularization constraints on attribution behavior during model training to improve both the rationality and robustness of attributions (Gao et al., 2024).

So far, improving the attribution rationality and invariance typically follows two distinct routes. A common method to enhance attribution rationality is to introduce human annotations (e.g., bounding boxes) as constraints to align attribution results with prior knowledge (Ross et al., 2017; Selvaraju et al., 2019; Yang et al., 2023). However, using human annotations as the criterion for correct attributions, in effect, forces the model's decision logic to align with human priors. This risks distorting the model's inherent reasoning mechanism and leading to erroneous attributions. Improving attribution invariance is generally achieved through self-supervised regularization strategies that enforce consistency of attributions under input transformations (Pillai & Pirsiavash, 2021; Pillai et al., 2022; Li et al., 2023). Most current methods rely on gradient-based attribution techniques (e.g., Grad-CAM (Selvaraju et al., 2020)) for parameter updates. However, such methods are known to suffer from low faithfulness that the explanations they produce may significantly deviate from the actual causal reasoning of the model. Enforcing these unreliable attributions as constraints during training could adversely affect model performance.

Based on the aforementioned issues, we choose a novel and highly faithful method, Submodular Subset Selection-based Attribution (SMDL) (Chen et al., 2024), to generate an explanation for the attribution consistency constraint. This approach predefines a set of submodular functions to evaluate attribution scores, modeling the attribution process as a continuous subset search procedure. By leveraging submodular functions and external constraints (including filtering high-confidence positive samples and limiting the area of attribution regions), it effectively excludes numerous decision-irrelevant areas. This allows the acquisition of high-quality, highly reliable attribution results without manual annotation, thereby avoiding the adverse effects of invalid attribution regions during training.

However, directly using the searched results as constraints faces two major challenges: First, as a black-box method, SMDL produces non-differentiable attributions, making it impossible to optimize model parameters directly via gradient updates. Second, although the final search results faithfully reflect the model's decision basis, they are generated through multi-step sequential iterations. Using only the final result as a constraint cannot guarantee the reproduction through the same path, thereby hindering effective enforcement of attribution consistency. To address this, we design a novel Submodular Ranking Loss based on the greedy search process of submodular selection. This loss utilizes the submodular function values as optimization response and employs a pairwise ranking loss to enforce that the corresponding transformed regions are selected in the order of the search path. Additionally, an extra constraint is imposed on the search termination, requiring that the consistency score of the last selected region exceeds a predefined stopping threshold. This guarantees

that the search process terminates at the same stage. The integration of these two components ultimately enables differentiable optimization of the black-box greedy search process. We conducted experimental validation across multiple dimensions, including task performance, attribution stability, faithfulness, rationality, and precision. The results demonstrate that using high-quality, highly faithful attribution results as regularization constraints significantly improves the model's attribution quality while minimizing the impact on task performance. The main contributions are as follows:

- We propose a self-supervised submodular searching regularization method. It constructs high-quality ground-truth through a high-faithfulness attribution mechanism and strict sample selection, enforcing geometric transformation consistency, thereby enhancing attribution rationality and stability.

- we propose a submodular ranking loss based on the greedy search process, which resolves non-differentiable and path-dependent issues encountered by submodular attribution, ensuring highly consistent final attribution results.

- Comprehensive evaluation across multiple metrics, including task performance, attribution stability, precision, faithfulness, and rationality, shows the proposed method significantly enhances attribution quality with minimal performance impact.

## 2 RELATED WORK

### 2.1 SEARCH-BASED ATTRIBUTION METHODS

The core idea is to generate high-precision attribution maps by executing an intelligent search strategy across the input space to locate the most critical local regions for the model prediction, optimized through an evaluation function. For example, SAGs (Karimi et al., 2023) employs a beam search algorithm to systematically explore multiple high-confidence local regions within an input image, thereby moving beyond the limitation of traditional saliency maps, which can only generate a single explanation. MoXI (Sumiyasu et al., 2024), a game theory-based attribution method, leverages Shapley values to identify critical pixel groups by quantifying their synergistic interactions, rather than evaluating each pixel contribution in isolation, thereby revealing the basis for model decisions with greater accuracy. SMDL (Chen et al., 2024) is an attribution method based on submodular optimization. It employs a search strategy to filter the most representative key regions from sparsified image patches according to multiple scoring criteria, thereby generating concise and interpretable attribution maps. VPS (Chen et al., 2025) is a novel attribution method based on visual precision search, specifically designed to interpret multimodal foundation models (e.g., Grounding DINO (Liu et al., 2024) and Florence-2 (Xiao et al., 2024)). By partitioning input images into sparse sub-regions and leveraging submodular functions for region selection, it generates highly precise attribution maps.

### 2.2 ATTRIBUTION REGULARIZATION

Attribution regularization (Gao et al., 2024) enhances model training by applying constraints directly to attribution maps, guiding the model to minimize reliance on noise and irrelevant features. This encourages the model to focus on meaningful patterns in the data, significantly improving the rationality and reliability of attribution results. Consistency is a core objective of attribution regularization, ensuring that the model's attribution results remain stable even when the input data distribution undergoes significant changes. Several methods have now been proposed to enhance the consistency of attribution regularization. For example, Pillai et al. (Pillai & Pirsiavash, 2021) proposed improving the consistency of model explanations by incorporating consistency constraints during training, resulting in more coherent and reliable explanations under a given attribution method. Li et al. proposed the DRE framework (Li et al., 2023), which leverages the differences between data distributions as a self-supervised signal to guide the model toward producing consistent and stable explanations for the same input, even under distribution shifts or environmental variations. This approach significantly enhances the robustness and reliability of model attributions in out-of-distribution (OOD) scenarios. Liu et al. proposed the ICEL (Liu et al., 2023) method, which introduces an Inconsistent Explanation Loss as its core component. This loss function optimizes the model by measuring the differences between explanation heatmaps corresponding to different predictions for the same image.

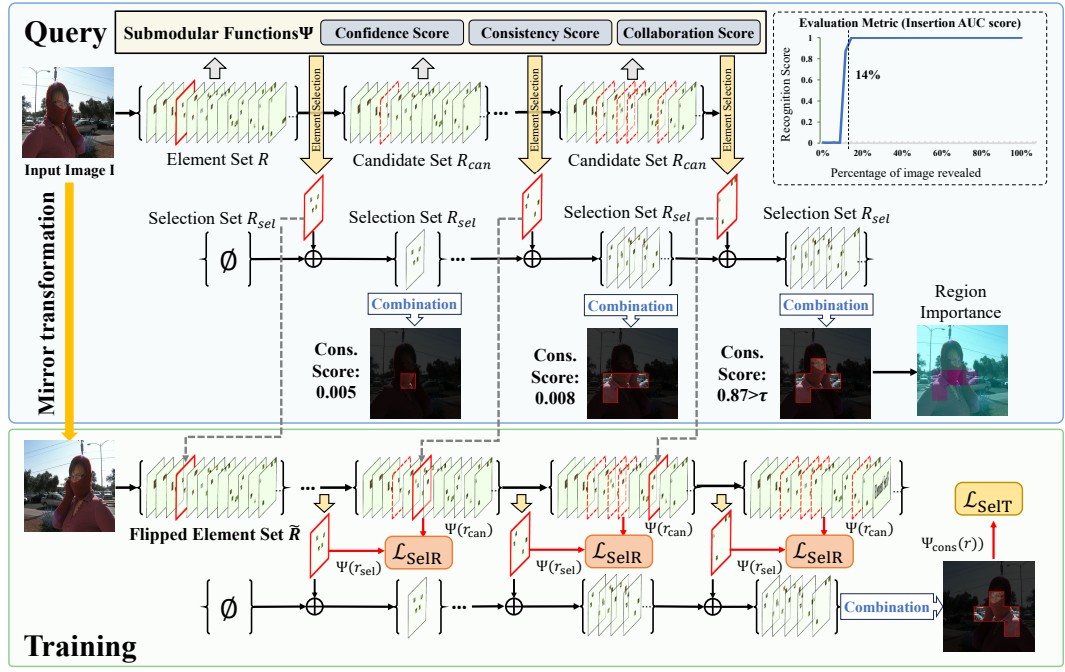

Figure 2: The complete computation of the Submodular Rank Loss with two stages. In the query stage, the original image is searched to obtain a decision region set $R_{\text{sel}}$, stopping when its consistency score meets a predefined threshold $\tau$. The sequence $R_{\text{sel}}$ serves as the supervision target. During training, a selection ranking loss $\mathcal{L}_{\text{SelR}}$ ensures each element in the path is properly selected from candidates, while an additional selection truncation loss $\mathcal{L}_{\text{SelT}}$ applied to the final element enforces consistent search termination.

## 3 METHOD

### 3.1 PRELIMINARY

In this section, we provide a brief introduction to the attribution method based on submodular subset selection, details are provided in the appendix B. The attribution search process could be reformulated as maximizing a submodular function $\Psi(S, \mathcal{F})$ measuring the interpretability of a selected subset of regions (Chen et al., 2024). A simple greedy algorithm is capable of identifying a near-optimal result as long as $\Psi$ is monotone non-decreasing. In this work, such a submodular function was proposed with three scoring components:

**Consistency score**. Ensuring that the semantic features of the selected region $S$ are consistent with the entire image $X$, which is denoted as:

$$\Psi_{\text{cons}}(S_k, \mathcal{F}) = \mathcal{F}(S_k)_{\bar{y}}, \tag{1}$$

where $\mathcal{F}$ represents the classification model to be explained, $\bar{y}$ is the predicted category, and the selected $S_k$ is derived by merging subsets of the currently selected regions $R_{\text{sel}} = \{r_1^*, \ldots, r_k^*\}$, denoted as $S_k = \textbf{SUM}(R_{\text{sel}})$.

**Confidence score**. Ensuring that the selected region $S$ enables the model to make predictions with a very high level of confidence, which is denoted as:

$$\Psi_{\text{conf}}(S_k, \mathcal{F}) = 1 - \frac{C}{\sum_{i=1}^{C}(\textbf{exp}(\mathcal{F}(\textbf{S}_\textbf{k})_\textbf{i}) + 1)}. \tag{2}$$

**Collaboration score**. Ensuring that upon covering the selected region $S$, the features of the remaining image parts are dissimilar to the entire image, which is denoted as:

$$\Psi_{\text{colla}}(S_k, X, \mathcal{F}) = 1 - \Psi_{\text{cons}}(X - S_k, \mathcal{F}). \tag{3}$$

The overall submodular function is calculated as:

$$\Psi = \lambda_1 \cdot \Psi_{\text{cons}} + \lambda_2 \cdot \Psi_{\text{conf}} + \lambda_3 \cdot \Psi_{\text{colla}}. \tag{4}$$

## 3.2 SUBMODULAR RANKING LOSS

Conventional attribution methods generate results in one step, whereas search-based methods produce explanations through a sequential decision process. Therefore, to enforce attribution invariance under geometric transformations, we must ensure that the greedy search path (i.e., the sequence of selected regions) remains consistent between original and transformed images, ultimately leading to identical final results. Motivated by this, we employ submodular function $\Psi(S, \mathcal{F})$ that measures the relevance of a region set $R$ to the parameters $\theta$ of model $\mathcal{F}$. Based on the greedy search trajectory, we propose a novel **Submodular Ranking Loss** that constraints the order of region selection to be invariant across transformations. The overall pipeline is illustrated in Fig. 3.

Given a training image and label $(X, y)$, we first obtain the attribution region query sequence for the image. The image is partitioned into $n$ sub-regions to form a candidate set $R_{\text{can}} = \{r_1, \cdots, r_n\}$ We then employ the SMDL method to perform attribution region search. The process terminates when the consistency score $\Psi_{\text{cons}}(S_k, \mathcal{F})$ of the currently selected subset reaches a predefined threshold $\tau_{\text{cons}}$ (set to $\tau_{\text{cons}} = 0.7$ in experiments). $S_k$ is taken as the final result, and the corresponding attribution region sequence $R_{sel} = \{r_1^*, \cdots, r_k^*\}$ is used as the supervision ground truth during training.

During the training phase, we constrain the model to ensure that the attribution search sequence under geometric transformations remains consistent with the query set $R_{\text{sel}}$. We adopt horizontal flipping to first obtain the flipped candidate region set $\tilde{R}_{\text{can}} = \{\tilde{r}_1, \cdots, \tilde{r}_n\}$ and the query region set $\tilde{R}_{\text{sel}} = \{\tilde{r}_1^*, \cdots, \tilde{r}_k^*\}$. At step $i$ of the search, given the current selected subset $\tilde{R}_i = \{\tilde{r}_1, \cdots, \tilde{r}_{i-1}\}$ and the candidate set $\tilde{R}_{\text{can}} \setminus \tilde{R}_i$, we enforce that the region $\tilde{r}_i^*$ should be chosen from the candidate set. This implies that the submodular function value of the new attribution region $\tilde{S}_i = \mathbf{SUM}(\tilde{R}_i) + \tilde{r}_i^*$ should be higher than that of any other candidate region. Since we only require that the region $\tilde{r}_i^*$ achieves the highest score rather than a strict total ordering, we formulate this **Selection Ranking loss** between $\tilde{r}_i^*$ and other candidate region $\tilde{r} \in \tilde{R}_{\text{can}} \setminus \tilde{R}_i$ using a pairwise ranking loss, i.e. $\mathbf{Relu}(\cdot)$:

$$\mathcal{L}_{\text{SelR}} = \frac{\sum\limits_{i=1}^{k} \sum\limits_{j=1}^{n-i} \mathbf{ReLU}(\Psi(S_{i-1} + \tilde{r}_j) - \Psi(S_{i-1} + \tilde{r}_i^*) + \delta)}{(2n - k - 1) \cdot k / 2}. \tag{5}$$

Here, $\delta$ is a parameter that controls the score margin between the selected region $\tilde{r}_i^*$ and other candidate regions $\tilde{r}$. When $i = 1$, $\tilde{R}_0 = \{\varnothing\}$ and $\tilde{S}_0$ serves as an all-zero starting point for the search. During the search process, we also set $\tau_{\text{cons}}$ as a stopping condition. Therefore, after the image is flipped, we require not only that the search trajectory remains consistent, but also that the stopping point remains the same, ensuring consistent explanation results. Thus, at the final step of the search, the consistency score of the selected region $\tilde{r}_k^*$ should exceed $\tau_{\text{cons}}$, ensuring that the attribution search process on the flipped image stops at this point. We define a **Selection Truncation loss** as follows:

$$\mathcal{L}_{\text{SelT}} = \mathbf{ReLU}(\tau_{\text{cons}} - \Psi_{\text{cons}}(S_{i-1} + \tilde{r}_k^*, \mathcal{F}) + \delta) \tag{6}$$

Based on the above content, we have constructed the final Submodular Ranking loss:

$$\mathcal{L}_{\text{SubR}} = \mathcal{L}_{\text{SelR}} + \mathcal{L}_{\text{SelT}}. \tag{7}$$

## 3.3 SUBMODULAR SEARCHING REGULARIZATION

We adopt a fine-tuning strategy based on pre-trained models to implement submodular searching regularization. This design is motivated by two key considerations: first, attribution search requires the model to already possess stable decision-making capabilities in order to effectively adjust its behavior based on attribution patterns; second, since the training process is self-supervised regularization learning, the quality of the query set $R_{\textbf{sel}}$ directly influences the final training outcome.

To ensure the quality of $R_{\textbf{sel}}$, we implement a two-tier strategy: first, regularization constraints are applied only to high-confidence correctly classified samples. Our objective is to enhance attribution

---

**Algorithm 1:** Submodular Searching Regularization

---

**Input:** Pretrained model $\mathcal{F}$ with parameter $\theta$, training set $\mathcal{D}$, searching truncation threshold $\tau_{cons}$, regularization strength $\lambda$, classification confidence threshold $\tau_{conf}$, searching area threshold $\tau_a$, learning rate $\alpha$.

**Output:** Finetuned Model $\mathcal{F}^*$ .

1 **for** *Epoch=1...N* **do**
2    **for** *training sample* $(X, y) \in \mathcal{D}$ **do**
3      Acquire the model prediction $z \leftarrow \mathcal{F}_\theta(X)$;
4      Divide image $X$ into region set $R = \{r_1, \cdots, r_n\}$;
5      Acquire attribution region sequence $R_{\text{sel}} = \{r_1^*, \ldots, r_k^*\} \leftarrow \textbf{SMDL}(\mathcal{F}, R, \tau_{\text{cons}})$;
6      Calculate classification loss $\mathcal{L}_{\text{cls}} \leftarrow \textbf{CE}(z, y)$;
7      **if** $\textbf{argmax}(z) = y$ *and* $\max(z) > \tau_{conf}$ *and* $\textbf{Area}(S) < \tau_a$ **then**
       `// Leveraging reliable sample for regularization`
8        $(\tilde{X}, \tilde{R}_{\text{sel}}) \leftarrow \textbf{HorizontalFlip}(X, R_{\text{sel}})$;
9        Calculate regularization based on 7 $\mathcal{L}_{\text{reg}} \leftarrow \mathcal{L}_{\text{SubR}}(\mathcal{F}, \tilde{X}, \tilde{R}_{\text{sel}})$;
10        $\mathcal{L} \leftarrow \mathcal{L}_{\text{cls}} + \lambda \cdot \mathcal{L}_{\text{reg}}$;
11      **else**
       `// Maintaining task constraints on unreliable sample`
12        $\mathcal{L} \leftarrow \mathcal{L}_{\text{cls}}$;
13      **end**
14      Update model parameters $\theta \leftarrow \theta - \eta \cdot \partial_\theta \mathcal{L}$
15    **end**
16 **end**

---

quality with minimal impact on task performance, as erroneous patterns in misclassified samples may adversely affect training. Second, we observe that directly applying SMDL attribution to pretrained models often includes excessive irrelevant regions (manifested as an overly large attribution area $\textbf{Area}(S)$, frequently exceeding two-thirds of the total image area). Since ideal attributions should be more focused and discriminative, we introduce an area threshold $\tau_a$ to select samples with more concentrated attributions for regularization training. The complete regularization workflow is summarized in Algorithm 1.

## 4 EXPERIMENT

### 4.1 EXPERIMENTAL SETTING

#### 4.1.1 EXPERIMENT SETUP

We conducted experiments using both ViT-Base/16 and ViT-Large/16 architectures on the ImageNet-100 dataset (Tian et al., 2020) to evaluate model performance. ImageNet100 is a subset of ImageNet-1K, comprising 100 selected categories. We utilize all training and validation images within these categories. Our approach was compared with the attribution regularization methods Grad-CAM Consistency (GC) (Pillai & Pirsiavash, 2021) and Contrastive Grad-CAM (CGC) (Pillai et al., 2022). We first fine-tuned the pre-trained model on the ImageNet100 dataset for 5 epochs to establish a baseline model. Both the GC and CGC methods were directly fine-tuned from the pre-trained model for 10 epochs. In contrast, our method involved an additional 5 epochs of fine-tuning on top of the baseline model.

#### 4.1.2 EVALUATION METRIC

We adopt a comprehensive set of metrics to evaluate both task performance and attribution quality. Classification performance is measured by Top-1 accuracy (Acc). Attribution quality is evaluated with several metrics. Stability is a modified Intersection-over-Union (IoU) that measures the overlap between attribution regions before and after image flipping, with an additional penalty term to account for excessively large attribution regions. Attribution Area (Attr. Area) quantifies the fraction of image pixels highlighted by the attribution map, reflecting the precision of explanations. We

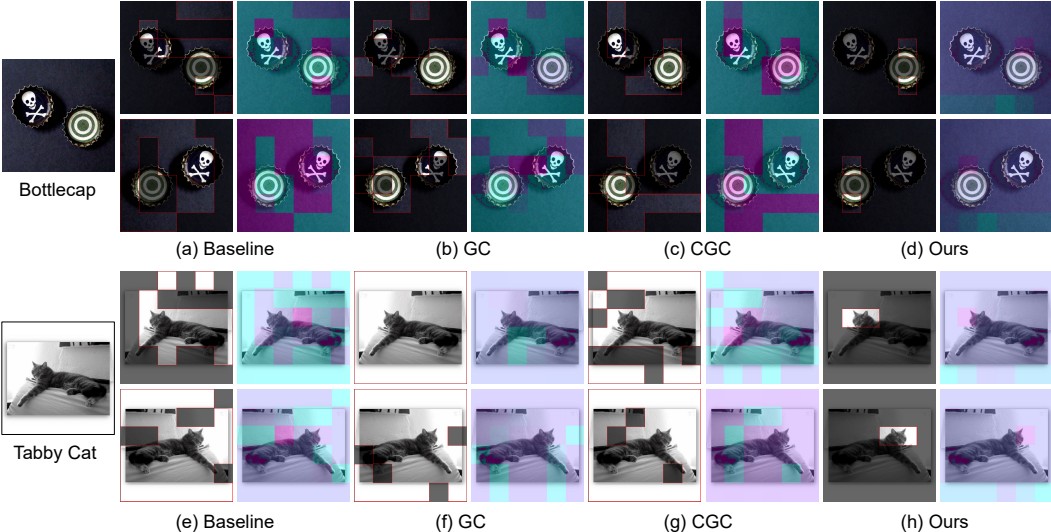

Figure 3: Comparison of attribution results and corresponding saliency maps before and after image flipping using different methods. The model employed is ViT-Base. (Best viewed when zoomed in.)

Table 1: Task performance and attribution quality metrics of across different methods. Faithfulness metrics PG are calculated via SDML.

| Model | Method | Acc (↑) | Stability (↑) | Insertion (↑) | Deletion (↓) | Attr. Area (↓) |
|-------|--------|---------|---------------|---------------|--------------|----------------|
| ViT-B/16 | Baseline | **0.9360** | 0.14 | 0.4496 | 0.1363 | 55.01% |
| | GC | 0.9194 | 0.14 | 0.4259 | 0.1002 | 58.99% |
| | CGC | 0.9260 | 0.13 | 0.4095 | 0.3127 | 59.49% |
| | Ours | 0.9332 | **0.27** | **0.6840** | **0.0515** | **24.48%** |
| ViT-L/16 | Baseline | **0.9218** | 0.15 | 0.4308 | 0.1396 | 56.92% |
| | GC | 0.8978 | 0.15 | 0.3910 | 0.0855 | 62.47% |
| | CGC | 0.9008 | 0.13 | 0.3859 | 0.1264 | 61.01% |
| | Ours | 0.9118 | **0.23** | **0.5494** | **0.0214** | **37.03%** |

also report Insertion and Deletion scores (Wang & Wang, 2024) to evaluate attribution faithfulness, where Insertion measures the increase in model confidence when the most relevant pixels are gradually inserted, and Deletion measures the decrease in confidence when the most relevant pixels are removed. Finally, Pointing Game (PG) (Nguyen et al., 2023) and Enhanced Pointing Game (EPG) (Gairola et al., 2025) are adopted to evaluate the spatial alignment between attribution maps and ground-truth object locations.

## 4.2 STABILITY

Table 1 reports the classification accuracy (Acc) and attribution stability of different models. We observe that both GC and CGC lead to non-trivial degradation in classification performance, with up to 2–3% accuracy drops compared to the baseline. In contrast, our method preserves the classification accuracy almost intact: on ViT-B/16, the accuracy decreases only marginally from 0.9360 (Baseline) to 0.9332, and on ViT-L/16 from 0.9218 to 0.9118, demonstrating that the primary task performance is largely unaffected. Meanwhile, our method yields substantial gains in Stability. For instance, on ViT-B/16, Stability nearly doubles from 0.14 (Baseline) to 0.27, while GC and CGC remain almost unchanged at 0.14 and 0.13, respectively. Similar improvements are observed for ViT-L/16, where Stability increases from 0.15 to 0.23. These results highlight that our submodular regularization enforces much stronger invariance of attribution maps under image transformations, without sacrificing classification accuracy.

Table 2: Rationality metrics of attribution for different methods. We evaluated both the black-box attribution method SMDL and the white-box attribution method Grad-Eclip. The best results in each column are highlighted in bold.

| Method | ViT-B/16 | | | | ViT-L/16 | | | |
|---|---|---|---|---|---|---|---|---|
| | $PG_{SMDL}$ | $EPG_{SMDL}$ | $PG_{Eclip}$ | $EPG_{Eclip}$ | $PG_{SMDL}$ | $EPG_{SMDL}$ | $PG_{Eclip}$ | $EPG_{Eclip}$ |
| Baseline | 0.8244 | 0.5508 | 0.8664 | 0.7400 | 0.8338 | 0.5553 | 0.4906 | 0.4479 |
| GC | 0.8246 | 0.5442 | **0.8840** | 0.7133 | 0.8392 | 0.5406 | 0.4982 | 0.4701 |
| CGC | 0.8278 | 0.5338 | 0.8544 | 0.6401 | 0.7990 | 0.5319 | **0.5630** | **0.5060** |
| Ours | **0.8906** | **0.6829** | 0.8638 | **0.7497** | **0.8626** | **0.6691** | 0.5594 | 0.5053 |

### 4.3 FAITHFULNESS

We next examine the faithfulness of attribution maps using the Insertion and Deletion metrics. Table 1 shows that our method consistently achieves the best trade-off between these two complementary measures. For ViT-B/16, the Insertion score increases from 0.4496 (Baseline) to 0.6840, while the Deletion score drops sharply from 0.1363 to 0.0515. Similar improvements are observed on ViT-L/16, where Insertion rises from 0.4308 to 0.5494 and Deletion decreases from 0.1396 to 0.0214. In comparison, existing regularization approaches (GC, CGC) often fail to deliver consistent gains. For example, CGC reduces Insertion to 0.4095 on ViT-B/16 and increases Deletion to 0.3127, suggesting that these methods may over-regularize and suppress informative features. Overall, the marked reduction in Deletion and the significant boost in Insertion highlight that our attributions are not only more compact but also more faithful to the decision-making process, yielding explanations that better align with human intuition.

### 4.4 RATIONALITY

We further assess rationality from two complementary perspectives: the compactness of attribution regions (Attr. Area) and the semantic alignment with target objects (PG/EPG). On Attr. Area, our method produces substantially more compact explanations. For instance, on ViT-B/16, the attribution area is reduced from 55.01% (Baseline) to 24.48%, and on ViT-L/16, from 56.92% to 37.03%. This demonstrates that our approach avoids covering irrelevant regions and instead focuses attribution on the most discriminative evidence. On PG and EPG, our method consistently outperforms all baselines. Specifically, on ViT-B/16, $PG_{SMDL}$ improves from 0.8244 to 0.8906, and $EPG_{SMDL}$ from 0.5508 to 0.6829. Similar gains are observed on ViT-L/16, where $PG_{SMDL}$ increases from 0.8338 to 0.8626, and $EPG_{SMDL}$ from 0.5553 to 0.6691. These results indicate that our attributions not only remain compact but also align more accurately with semantically meaningful object regions. In contrast, existing regularization baselines (GC, CGC) exhibit instability: while sometimes improving one metric, they often lead to enlarged attribution regions (e.g., CGC exceeding 59% area on ViT-B/16) and degraded PG/EPG performance. Our method, achieves improvements on both compactness and semantic rationality simultaneously, yielding more interpretable explanations.

### 4.5 ABLATION STUDY

#### 4.5.1 SUBMODULAR FUNCTION

We conduct ablation study to verify the impact of different submodular scores or combinations of submodular scores on model performance and attribution quality by ablating different terms of the submodular function. The experiments were conducted on the ViT-B model. During training with different ablated terms, the weights of all submodular score terms were set to 1.0. The experimental results are shown in Table 4.5.1. As can be seen from the results, the model achieves the best performance across all evaluation metrics when all three submodular scores are used for training. The second tier of methods includes the combination of Conf. + Colla. and Confidence alone, which perform close to the best, even surpassing the full version by 0.0308 in terms of attribution rationality (Conf. + Colla.). Following these are Cons. + Colla. and Collaboration alone, which show attribution metrics similar to those of the pre-trained model but with a decrease in accuracy.

Table 3: Task performance and attribution quality metrics of ViT-B model under different combinations of submodular function components. Location metrics PG are calculated via SDML.

| Submodular Function | | | Acc. (↑) | Stab. (↑) | Area (↓) | Insert. (↑) | PG (↑) |
|---|---|---|---|---|---|---|---|
| Conf. Score | Cons. Score | Colla. Score | | | | | |
| ✔ | ✗ | ✗ | 0.9320 | 0.25 | 55.02% | 0.4620 | 0.8366 |
| ✗ | ✔ | ✗ | 0.9318 | 0.09 | 74.00% | 0.2778 | 0.9124 |
| ✗ | ✗ | ✔ | 0.9306 | 0.12 | 58.04% | 0.4001 | 0.8730 |
| ✔ | ✔ | ✗ | 0.9332 | 0.05 | 80.78% | 0.2064 | 0.9220 |
| ✔ | ✗ | ✔ | 0.9288 | 0.26 | 32.85% | 0.6661 | 0.9214 |
| ✗ | ✔ | ✔ | 0.9252 | 0.16 | 46.79% | 0.4543 | 0.8360 |
| ✔ | ✔ | ✔ | 0.9332 | 0.27 | 37.03% | 0.6840 | 0.8906 |

Table 4: Task performance and attribution quality metrics of ViT-B model under different weights of attribution regularization constraints. Location metrics PG and EPG are calculated via SDML.

| Reg. Weight $\lambda$ | Acc (↑) | Stab. (↑) | Inser. (↑) | Delet. (↓) | Area (↓) | PG (↑) | EPG (↑) |
|---|---|---|---|---|---|---|---|
| 0 | 0.9360 | 0.14 | 0.4496 | 0.1363 | 55.01% | 0.8244 | 0.5508 |
| 0.01 | 0.9346 | 0.29 | 0.7050 | 0.0496 | 24.31% | 0.8744 | 0.6788 |
| 0.10 | 0.9332 | 0.27 | 0.6840 | 0.0515 | 24.48% | 0.8906 | 0.6829 |
| 0.50 | 0.9316 | 0.27 | 0.6409 | 0.0229 | 29.88% | 0.8840 | 0.6641 |
| 1.00 | 0.9282 | 0.34 | 0.7063 | 0.1580 | 28.99% | 0.9172 | 0.7104 |

### 4.5.2 REGULARIZATION STRENGTH $\lambda$

We systematically evaluate the impact of the attribution regularization weight ($\lambda$) on the task performance and attribution quality of the ViT-B model, as reported in Table 4. When $\lambda$ increases from 0 to 1.00, classification accuracy (Acc) experiences only a marginal decrease from 0.9360 to 0.9282, confirming that the primary task performance is well-maintained. In contrast, attribution metrics show substantial improvements. Most notably, the Deletion (↓) score plunges from 0.1363 ($\lambda = 0$) to 0.0229 ($\lambda = 0.50$), indicating that the explanations become far more faithful by focusing on truly relevant features. The Insertion (↑) score also improves significantly, from 0.4496 to over 0.7063 for $\lambda \geq 0.01$. The weight $\lambda = 0.10$ appears to be a favorable balance point, where attribution quality metrics like PG (↑) and EPG (↑) see significant gains (e.g., PG: $0.8244 \rightarrow 0.8906$) with a negligible accuracy drop (less than 0.003). Excessively high weights (e.g., $\lambda = 1.00$) may improve stability but can introduce instability in other metrics. The results validate that the choice of $\lambda$ provides a tunable knob to balance between task accuracy and explanation quality, with an intermediate range ($\lambda = 0.01 \sim 0.50$) offering the most favorable trade-off.

## 5 CONCLUSION

This paper proposed a training framework based on high-fidelity submodular attribution to enhance the stability and rationality of model attributions. To improve attribution rationality, the proposed method leverages high-fidelity submodular attribution to automatically extract compact and discriminative pseudo-ground-truth regions, providing high-quality, annotation-free supervision for training. To address the non-differentiability and path-dependency of submodular search, a submodular ranking loss is proposed, which enforces consistency in search paths and alignment of their termination under geometric transformations, thereby achieving differentiable optimization of the greedy search procedure. Results across five key dimensions (classification accuracy, attribution stability, faithfulness, rationality, precision) verify that the framework enables models to learn more reliable and consistent explanations without human supervision.

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

## A APPENDIX

### A.1 USE OF LLM

**Declaration:** The use of DeepSeek in the preparation of this manuscript was strictly limited to grammatical improvements and text polishing.

## B INTRODUCTION OF SUBMODULAR SELECTION

A submodular function is a set function with the property of diminishing marginal returns. A submodular function is a set function defined on a finite set $S$, denoted as $\Psi : 2^S \to \mathbb{R}$. For any subsets $S_a \subseteq S_b \subseteq S$ any elements $s \in S \setminus S_b$, it satisfies the following condition:

$$\Psi(S_a \cup \{s\}) - \Psi(S_a) \geq \Psi(S_b \cup \{s\}) - \Psi(S_b). \tag{8}$$

This property is also described as the characteristic of "diminishing marginal returns". We then review the process of attribution region search. We aim to identify the most critical regions that support the model's decision from the set of subregions partitioned from the image. This means that the initial search results should have a higher contribution to the model's decision, whereas the marginal contribution of regions added later diminishes. Therefore, the attribution search process inherently

aligns with the property of submodular function and could be reformulated as maximizing a submodular function $\Psi(S)$ measuring the interpretability of a selected subset of regions. The primary advantage of this approach lies in the fact that, provided a monotone non-decreasing submodular function can be constructed, a simple greedy algorithm is capable of identifying a near-optimal subset within a large search space (Nemhauser et al., 1978).

