# OpenReview forum: "Subset Selection-based Attribution Regularization for Rational and Stable Interpretability"
_ICLR.cc/2026/Conference — ICLR 2026 Conference Withdrawn Submission_

### Official Review · Reviewer_JV61 · 2025-10-28

**Soundness:** 2
**Presentation:** 2
**Contribution:** 2
**Rating:** 4
**Confidence:** 5

**Summary:**

This paper proposes a submodular selection–based attribution regularization method aimed at improving the rationality and stability of model interpretability. By defining a submodular function and a ranking(SubR) loss , the authors introduce a high-fidelity attribution constraint into the training process, enabling self-supervised consistency learning for explanations. The overall framework is conceptually novel, and experimental results demonstrate clear advantages over traditional baselines in multiple attribution quality metrics. However, the paper still suffers from notable deficiencies in notation clarity, experimental design, and completeness of comparative studies.

**Strengths:**

1\.The combination of submodular search and differentiable ranking loss offers a new perspective for addressing the non-differentiability issue in black-box attribution.

2\.The paper maintains a coherent logical flow from methodology to experiments, with a reasonably complete set of evaluation metrics (including stability, faithfulness, and rationality).

3\.The framework avoids reliance on human annotations, which enhances its generality and potential applicability.

**Weaknesses:**

1\.Ambiguous notation and unclear mathematical definitions – In Equations (1)–(3), the meanings of the symbols  S\_k 、R\_sel and SUM(·) are unclear. For instance, S\_k is described as both a merged region set derived from R\_sel and as a “region,” but its exact form—matrix, tensor, or set—is never specified. The operation “SUM” in Equation (1) is also undefined: does it denote pixel summation, averaging, or a logical union? Such ambiguity severely undermines reproducibility. Furthermore, S\_k is not boldfaced in Equation (1) but is boldfaced in Equation (2); it remains unclear whether they represent the same variable. Consistency and explicit clarification are needed.

2\.Limited experimental design and outdated baselines – The experiments are conducted only on ImageNet-100 using the ViT architecture, without validation on larger-scale datasets (e.g., ImageNet-1K) or mainstream architectures (e.g., ResNet, ConvNeXt). This limitation prevents a convincing demonstration of generalizability. In addition, the comparative baselines (GC and CGC) are both Grad-CAM–based and therefore of relatively low faithfulness. It is recommended to include more recent and representative attribution regularization approaches, such as attention-based or high-fidelity attribution–guided learning methods.

3\.Insufficient result presentation and analysis – The ablation study does not convincingly demonstrate the necessity of each component. In the full model evaluation, two out of five metrics are not optimal, and classification accuracy is not significantly higher than the baseline, weakening the claimed superiority. Moreover, the (Conf. + Colla.) combination even surpasses the full version in certain metrics, challenging the “all-three-components-are-essential” assertion. There are also presentation issues: some figures (e.g., Figure 2) are not referenced or explained in the main text, weakening figure–text alignment. Additionally, in Table 3, the best results should be highlighted in bold and the second-best underlined to improve readability and comparability.

**Questions:**

Please address the questions raised in the Weaknesses section.

---

### Official Review · Reviewer_hqUc · 2025-10-31

**Soundness:** 2
**Presentation:** 3
**Contribution:** 2
**Rating:** 4
**Confidence:** 4

**Summary:**

This paper proposes a training framework based on high-faithfulness submodular attribution to enhance the stability and rationality of model explanations. The authors argue that existing methods are limited, as supervised approaches rely on human priors that may conflict with the model's logic, while self-supervised methods often use low-faithfulness attributions (e.g., Grad-CAM), leading to misaligned explanations. To address this, the paper employs the SMDL attribution method to automatically extract pseudo-ground-truth regions. A novel "Submodular Ranking Loss" is introduced to overcome the non-differentiable and path-dependent nature of the SMDL search, enforcing attribution consistency under geometric transformations. Experiments demonstrate that this method improves attribution quality with minimal impact on classification accuracy.

**Strengths:**

1.  The paper addresses a clear and important problem in XAI: improving the stability and rationality of explanations, rather than just focusing on faithfulness.
2.  The idea of using a high-faithfulness attribution method (SMDL) to generate self-supervised signals, rather than relying on human annotations or low-faithfulness gradients, is a valuable contribution.
3.  The proposed Submodular Ranking Loss is a novel technique for optimizing a black-box, non-differentiable greedy search process, which is a non-trivial technical challenge.
4.  The experimental evaluation is comprehensive, covering task performance, stability, faithfulness (Insertion/Deletion), and rationality (PG/EPG).

**Weaknesses:**

1.  The method's design appears to be overly intuitive and lacks rigorous theoretical guarantees.
2.  The greedy algorithm relies on monotonicity. However, judging from the insertion curves generated by almost all current explainability algorithms, this property does not seem to hold in practice (as almost every curve shows a drop in confidence after adding certain features).
3.  The loss function is biased towards increasing confidence. However, what if the model's current confidence is the correct or intended one? A higher confidence score is not necessarily better for an explanation.
4.  A discussion on computational complexity is missing. This means it is difficult to confirm the method's performance and scalability on more complex models.
5.  The explanation goal is focused on the fine-tuned model. What if I want to obtain the explanation results for the original model? Is there any way to prove that this explanation is definitely applicable to the original model, for instance, by proving that the error in the explainability metric has an acceptable lower bound under a certain level of parameter intervention?

**Questions:**

1.  Your regularization method fine-tunes a pre-trained model. This implies the final explanations are for the *fine-tuned* model. If my objective is to explain the *original* pre-trained model, is this method applicable? Can you provide any theoretical bounds or guarantees on how the explanations from the fine-tuned model relate to the original one?
2.  The SMDL method and the greedy algorithm depend on the monotonicity of the submodular function. As noted in Weakness #2, insertion curves are often non-monotonic. How do you justify that the weighted sum of the three submodular scores ($\Psi_{cons}$, $\Psi_{conf}$, $\Psi_{colla}$) indeed satisfies the monotone non-decreasing property required for the greedy algorithm?
3.  What is the computational overhead of this method? The process requires executing the SMDL search to get $R_{sel}$ and then computing $\mathcal{L}_{Se1R}$ for each sample during training. Please provide a complexity analysis.

---

### Official Review · Reviewer_1KgZ · 2025-10-31

**Soundness:** 3
**Presentation:** 2
**Contribution:** 2
**Rating:** 4
**Confidence:** 4

**Summary:**

This paper introduces a self-supervised regularization framework to improve the stability and rationality of attributions in deep learning models. The core of the framework is the use of the SMDL attribution method to generate pseudo-labels and a novel "Submodular Ranking Loss" to handle the non-differentiable nature of the SMDL search. This loss function enforces consistency in the attribution search path under geometric transformations. Experiments on ImageNet-100 show that the proposed method significantly improves attribution stability and faithfulness scores compared to baselines without a significant drop in classification accuracy.

**Strengths:**

1.  The motivation is clear and correctly identifies the shortcomings of existing attribution regularization methods that rely on either human priors or low-faithfulness gradients.
2.  The translation of the black-box SMDL search process into a differentiable training loss is a clever design.
3.  The quantitative results are strong, showing clear improvements in stability, Insertion, and Deletion scores while maintaining high task performance.
4.  The method's ability to improve explainability with minimal impact on the model's Top-1 accuracy is a significant practical advantage.

**Weaknesses:**

1.  The number of training epochs for different methods is inconsistent. This requires a stricter discussion, including the training epochs before fine-tuning.
2.  The paper lacks comparisons against some obvious and important methods, for example, RISE.
3.  The baseline used for the insertion and deletion metrics needs to be discussed. Furthermore, it would be better to include more advanced metrics (e.g. ROAD [1]).
4.  The granularity of the region segmentation is an implicit hyperparameter that requires discussion.
5.  The use of hyperparameters and the resulting "smaller area" of attributions appear to be highly coupled.

Reference

[1] Rong, Yao, et al. "Evaluating feature attribution: An information-theoretic perspective." International Conference on Machine Learning. 2022.

**Questions:**

1.  Baseline model is fine-tuned for 5 epochs, while GC and CGC are trained for 10 epochs. The proposed method ("Ours") is trained for an additional 5 epochs on top of the 5-epoch baseline. This makes the total epochs (5 vs. 10 vs. 5+5) and the starting points inconsistent. Could you justify this experimental setup? Does this not create an unfair comparison?
2.  What baseline (e.g., blurred, grey, or random noise) was used when calculating the Insertion and Deletion scores?
3. Have you considered using or comparing against more advanced faithfulness metrics, such as ROAD, given the known limitations of Insertion/Deletion?
4. The SMDL method requires partitioning the image into a set of regions $R$. The granularity of this partitioning (e.g., patch size, number of patches) seems to be a critical implicit hyperparameter, but it is not discussed. How was this granularity chosen, and how sensitive are the results to it?
5. Algorithm 1 introduces an area threshold $\tau_{a}$ to select samples with more concentrated attributions for regularization. The results show a significant reduction in "Attr. Area". To what extent is this reduction a result of the $\tau_{a}$ filtering versus the $\mathcal{L}_{SubR}$ regularization itself?

---

### Official Review · Reviewer_pBeH · 2025-10-31

**Soundness:** 2
**Presentation:** 3
**Contribution:** 2
**Rating:** 4
**Confidence:** 4

**Summary:**

This paper proposes a training framework to improve the stability and rationality of model attributions. The authors use a high-faithfulness submodular attribution method to generate self-supervised signals. A novel ranking loss is designed to enforce consistency of the attribution search path under a geometric transformation. The experimental results suggest that this method enhances attribution quality metrics, such as stability, while maintaining task accuracy.

**Strengths:**

1.  The paper attempts to solve a challenging and practical problem in XAI: ensuring that explanations are not only faithful but also stable and consistent for semantically similar inputs.
2.  The self-supervised, annotation-free approach, which bootstraps from a high-faithfulness attribution method, is a promising direction for scalability.
3.  The visualizations in Figure 1 and Figure 3 intuitively demonstrate that the regularized model produces more consistent and compact attributions after flipping compared to the baseline.

**Weaknesses:**

1.  Humans sometimes also exhibit perceptual biases due to flipping (e.g., many visual illusions). Why do we need to emphasize that the explainability algorithm's explanation must be corresponding after a flip? Explainability algorithms need to be responsible for many unexpected and accidental situations, just as insurance is for unexpected events. When a model has a problem, we hope to use explainability algorithms to find the problem. This needs a more in-depth discussion, as I do not think this is a reasonable evaluation metric or algorithm design.
2.  The evaluation system and the method design (SMDL) are too highly coupled. This means the evaluation will inherently favor the method itself, rather than true faithfulness.
3.  Other transformations besides horizontal flipping also need to be considered (at least from the evaluation perspective).
4.  Regarding the experimental results, my current feeling is that they are insufficient. The paper mentions many attractive domains where this could be used but provides no validation. However, I do not think the author should add these experiments, as this is not the basis for my decision, and current explainability work is often designed this way.

**Questions:**

1.  The core design of the paper enforces attribution consistency under horizontal flipping. As noted in Weakness #1, human perception itself is not perfectly invariant to such transformations. Can you provide a stronger justification for why "flip-invariance" is a necessary and rational design goal for an XAI algorithm, rather than a potentially overly-restrictive constraint that might mask the model's true behavior?
2.  The stability evaluation appears to be limited to horizontal flipping. Was the attribution stability tested against other common transformations, such as rotation, scaling, or brightness changes?

---

### Note · Authors · 2025-11-18

I have read and agree with the venue's withdrawal policy on behalf of myself and my co-authors.